

# Inhibition of non-small cell lung cancer (NSCLC) proliferation through targeting G6PD

Makamas Chanda[1,*], Pornchai Anuntasomboon[2,*], Komkrit Ruangritchankul[3], Poonlarp Cheepsunthorn[4] and Chalisa L. Cheepsunthorn[5]

[1] Interdisciplinary Program of Biomedical Sciences, Graduate School, Chulalongkorn University, Bangkok, Thailand
[2] Medical Sciences Program, Faculty of Medicine, Chulalongkorn University, Bangkok, Thailand
[3] Department of Pathology, Faculty of Medicine, Chulalongkorn University, Bangkok, Thailand
[4] Department of Anatomy, Faculty of Medicine, Chulalongkorn University, Bangkok, Thailand
[5] Department of Biochemistry, Faculty of Medicine, Chulalongkorn University, Bangkok, Thailand
[*] These authors contributed equally to this work.

Corresponding author
Chalisa L. Cheepsunthorn, chalisa.l@chula.ac.th

## ABSTRACT

**Background**. Mounting evidence has linked cancer metabolic reprogramming with altered redox homeostasis. The pentose phosphate pathway (PPP) is one of the key metabolism-related pathways that has been enhanced to promote cancer growth. The glucose 6-phosphate dehydrogenase (G6PD) of this pathway generates reduced nicotinamide adenine dinucleotide phosphate (NADPH), which is essential for controlling cellular redox homeostasis.

**Objective**. This research aimed to investigate the growth-promoting effects of G6PD in non-small cell lung cancer (NSCLC).

**Methods**. Clinical characteristics and G6PD expression levels in lung tissues of 64 patients diagnosed with lung cancer at the King Chulalongkorn Memorial Hospital (Bangkok, Thailand) during 2009-2014 were analyzed. G6PD activity in NSCLC cell lines, including NCI-H1975 and NCI-H292, was experimentally inhibited using DHEA and siG6PD to study cancer cell proliferation and migration.

**Results**. The positive expression of G6PD in NSCLC tissues was detected by immunohistochemical staining and was found to be associated with squamous cells. G6PD expression levels and activity also coincided with the proliferation rate of NSCLC cell lines. Suppression of G6PD-induced apoptosis in NSCLC cell lines by increasing Bax/Bcl-2 ratio expression. The addition of D-(-)-ribose, which is an end-product of the PPP, increased the survival of G6PD-deficient NSCLC cell lines.

**Conclusion**. Collectively, these findings demonstrated that G6PD might play an important role in the carcinogenesis of NSCLC. Inhibition of G6PD might provide a therapeutic strategy for the treatment of NSCLC.

## INTRODUCTION

Lung cancer accounted for the deaths of over 2 million people globally in 2018 (*Bray et al., 2018*). Non-small cell lung cancer (NSCLC) is the main type, representing approximately 80–85% of lung cancer. If the cancer is detected early or before metastasis, the 5-year survival rate for NSCLC is predicted to be around 60% (*Ehrenstein et al., 2023*). Nonetheless, the average survival time drops to less than 12% without treatment (*Wao et al., 2013*). Small-cell lung cancer (SCLC) is another less common but more aggressive type of lung cancer than NSCLC (*Bernhardt & Jalal, 2016*). It would be beneficial to have a better understanding of how lung cancer cells reprogram their metabolic pathways, since this knowledge may help in the early diagnosis of lung cancer.

Energy metabolic reprogramming (EMR) and altered redox homeostasis are emerging hallmarks of cancer (*De Santis et al., 2018*; *Xing et al., 2022*). In addition to aerobic glycolysis, or the Warburg effect, the pentose phosphate pathway (PPP), a branch of glycolysis, is crucial for tumorigenesis (*Xing et al., 2022*). PPP is composed of the oxidative branch and the non-oxidative branch (*Frederiks et al., 2008*). The oxidative branch, in which glucose 6-phosphate dehydrogenase (G6PD) is a rate-limiting enzyme, generates reduced nicotinamide adenine dinucleotide phosphate (NADPH) and ribose 5-phosphate (R5P), which are necessary for controlling redox homeostasis, the biosynthesis of fatty acids, and serving as a key component in the synthesis of nucleotides (*Benito et al., 2017*; *Sreedhar & Zhao, 2018*; *Yang, Stern & Chiu, 2021*). On the other hand, the non-oxidative branch produces fructose 6-phosphare (F6P), glyceraldehyde 3-phosphate (G3P), and pentose phosphates, which are supplements for glycolysis and anabolic pathways (*Frederiks et al., 2008*). Upregulation of G6PD has been reported to be associated with metastasis, advanced stage, and poor overall survival time (*Benito et al., 2017*; *Sreedhar & Zhao, 2018*; *Yang, Stern & Chiu, 2021*) in many malignancies, including colorectal cancer (*Ju et al., 2017*), bladder cancer (*Zhang et al., 2017b*), breast cancer (*Benito et al., 2017*; *Pu et al., 2015*), clear cell renal cell carcinoma (*Zhang et al., 2017a*), and lung cancer (*Hong et al., 2018*). A lower survival rate was observed in lung cancer patients with G6PD-positive cells compared to G6PD-deficiency patients (*Nagashio et al., 2019*). However, there was no report on the difference between G6PD expression in NSCLC and SCLC tissues. The effect of G6PD expression level on the proliferation of lung cancer is still unclear.

In this study, we aimed to examine the expression of G6PD in lung tissues from patients diagnosed with lung cancer. We also aimed to elucidate the growth-promoting effects of G6PD in NSCLC cell lines.

## MATERIALS & METHODS

### Tissue specimens

This study was approved by the Institute Ethics Committee of the Faculty of Medicine, Chulalongkorn University, Bangkok, Thailand (IRB 561/59, COA No. 1034/2016). The study protocol was performed according to the declaration of Helsinki for the participation of human individuals. Lung tissue specimens from biopsies histologically examined for diagnosis prior to treatment were obtained from 64 lung cancer patients who were admitted

to the King Chulalongkorn Memorial Hospital (Bangkok, Thailand) during 2009–2014. All tissue samples were examined by a team of pathologists at the King Chulalongkorn Memorial Hospital to determine the type and stage of cancer. Clinical data were collected at the time of the first diagnosis and continued until recurrence, death, or the last follow-up appointment.

## Histological and immunohistochemical analyses

Formalin-fixed and paraffin-embedded lung tissue blocks were serially cut into 5 $\mu$m thick sections, mounted on slides in serial order and processed following standard procedures for histological evaluation. Hematoxylin and eosin (H&E) staining was performed to determine the cancerous and adjacent non-cancerous areas on the sections. To detect G6PD protein expression, tissue sections adjacent (serial) to H&E-stained sections were heated in an autoclave at 120 °C for 10 min for antigen retrieval, treated with hydrogen peroxide to quench endogenous peroxidase, and blocked with corresponding serum from a secondary antibody raised. Subsequently, the sections were incubated with an anti-G6PD antibody produced in rabbits (HPA000247; Sigma-Aldrich, St. Louis, MO, USA) at a dilution of 1:1000 as the manufacturer's recommendation at 4 °C overnight. Detection was performed using a biotinylated secondary antibody (Sigma-Aldrich) followed by a streptavidin-biotin complex peroxidase (1:200; Vector, Burlingame, CA, USA) and visualized with 3,3′-diaminobenzidine tetrahydrochloride (Sigma-Aldrich). An isotype IgG antibody was used as a negative control. No counterstain was used. According to data from the Human Protein Atlas database (https://www.proteinatlas.org/ENSG00000160211-G6PD/tissue), G6PD exhibits high expression in testis tissue, which was utilized as the positive control. The stained sections were evaluated by two pathologists who were blinded to the patients' clinical information. The G6PD H-score was calculated using the formula 1× (% of weakly-stained as light brown, 1+) +2× (% of moderately stained as medium brown, 2+) +3× (% of strongly stained as dark brown, 3+), as described previously (*Leesutipornchai et al., 2020*).

## Cell culture and treatments

Beas-2B (epithelial cells isolated from normal human bronchial epithelium; ATCC# CRL-3588) and human NSCLC cell lines, including NCI-H1975 (lung epithelial cells derived from adenocarcinoma tissue; ATCC# CRL-5908), and NCI-H292 (human lung mucoepidermoid carcinoma cells; ATCC# CRL-1848), were obtained from the American Type Culture Collection (ATCC, Manassas, VA, USA). Beas-2B were cultured in DMEM, whereas NCI-H1975 and NCI-H292 were cultured and routinely passaged in RPMI-1640 medium containing 10% heat inactivated FBS and 1% penicillin/streptomycin solution (Merck Millipore, MA, USA) at 37 °C in 5% $CO_2$. Dehydroepiandrosterone (DHEA; Sigma-Aldrich) was prepared as a 1000-fold stock solution in dimethyl sulfoxide (DMSO), thus the final concentration of DMSO did not exceed 0.1%. D-(-)-ribose (R9629; Sigma-Aldrich) was dissolved in culture medium and sterilized by filtration through a 0.45 $\mu$m filter before use. After treatment with DHEA for 48 h, the cells were used as indicated in each experiment.

### siG6PD and transfection assay

Cells were plated in 12-well plates at 30–50% confluency in complete medium 24 h before transfection. Transfection was performed using Lipofectamine® 3000 (ThermoFisher Scientific, Waltham, MA, USA), following the manufacturer's instructions. The siG6PD for G6PD (sense sequence: GGCCGUCACCAAGAACAUU) and non-silencing scramble (sense sequence: GGCACUACCAGACACGAUU) were synthesized and purified by Integrated DNA Technologies, Inc., (IDT, Coralville, IA. USA) and were used at a 100 nM final concentration. After transfection of siG6PD for 24 h, the cells were used in the indicated experiments.

### Cell proliferation assay

After treatment, cells were incubated with MTT tetrazolium salt (final concentration 0.5 mg/ml, Sigma-Aldrich) for 2 h at 37 °C. The formazan crystal product was then dissolved with DMSO, and the optical absorbance was measured at 570 nm using a Synergy HT microplate reader (BioTek Instruments Inc., Winooski, VT, USA).

### Clonogenic assay

Cells were seeded in a 6-well plate at a density of $2 \times 10^2$ cells/well for 24 h before beginning treatment. After treatment, the culture medium was removed and replaced with a fresh complete medium. The cells were then continuously cultured for 7 days. At the end, the colonies were washed with 1X PBS, fixed with glutaraldehyde (0.6% v/v), stained with crystal violet (0.5% w/v) for 30 min, and counted.

### G6PD activity assay

After treatments, the cell pellets were collected at the indicated times and washed with ice-cold PBS. The cells were resuspended in cold PBS, sonicated for 10 s (repeated three times), and cooled on ice. Total protein was determined by the BCA protein assay kit (Thermo Fisher Scientific, Waltham, MA, USA), according to the manufacturer's instructions. G6PD enzyme activity was measured as previously described (*Garcia-Nogales et al., 1999*) with some modifications. Briefly, 20 μl of the cell lysate (protein at 1mg/ml concentration) was mixed with 980 μl of the reaction buffer, containing 0.38 mM NADP, 6.3 mM $MgCl_2$, 3.3 mM glucose 6-phosphate, and 5 mM maleimide in 50 mM Tris–HCl (pH 7.5) buffer. The absorbance was kinetically measured at 340 nm for 15 min at 37 °C using a Synergy HT microplate reader (BioTek Instruments Inc.). Enzyme activity was calculated using a standard curve of NADPH and expressed as NADPH units/min/mg of total protein.

### Scratch wound assay

Cells were seeded in 24-well plates to a final density of $1 \times 10^5$ cells/well and maintained in a $CO_2$ incubator at 37 °C for 24 h to allow cell adhesion. The confluent monolayer was scratched with a sterile 200-μl pipette tip. Then, culture medium containing dislodged cells was immediately removed and replaced with fresh medium, either alone or containing DHEA or siG6PD. The scratched areas were monitored by collecting digitized images at various time points or until closure of the wound in the control monolayer.

## Quantitative RT-PCR

Total RNA was isolated from cell pellets using TRIzol® Reagent (Thermo Fisher), according to the manufacturer's instructions. The quality and concentration of RNA were determined using a NanoDrop 2000 spectrophotometer (Thermo Fisher Scientific). cDNAs were synthesized using a RevertAid First Strand cDNA Synthesis Kit (Thermo Fisher Scientific), according to the manufacturer's instructions. Gene expression analysis was performed using PowerUp™ SYBR® Green Master Mix (Thermo Fisher Scientific) on a StepOnePlus real-time PCR machine (Thermo Fisher Scientific), according to the manufacturer's protocol. Primer sequences were as follows: G6PD forward primer 5′-GTC AAGGTGTTGAAATGCATC-3′ and reverse primer 5′-CATCCCACCTCTCATTCTCC-3′, Bax forward primer 5′-AACATGGAGCTGCAGAGGAT-3′ and reverse primer 5′- CAGC CCATGATGGTTCTGAT-3′, Bcl-2 forward primer 5′-GGTGGGGTCATGTGTGTG-3′ and reverse primer 5′-CGGTTCAGGTACTCAGTCATC-3′, and ß-actin as a reference gene forward primer 5′-ACTCTTCCAGCCTTCCTTC-3′ and reverse primer 5′-ATCTCCTTC TGCATCCTGTC-3′. The relative abundance of each target gene was calculated relative to ß-actin. The fold change in expression levels was reported as $2^{-\Delta\Delta Ct}$.

## Statistical analysis

Statistical analyses were performed using SPSS v.22.0 (SPSS Inc., Chicago, IL, USA). Data were obtained from three independent experiments performed in triplicate and presented as the mean $\pm$ standard deviation (SD). The chi-square test ($\chi^2$) and Student's $t$-test were used to examine the differences between categorical and quantitative variables. A two-sided difference with a $p$-value less than 0.05 was considered statistically significant.

# RESULTS

## Characteristics of patients

A total of 64 lung cancer patients, consisting of 44 NSCLC patients (68.75%) and 20 SCLC patients (31.25%), were examined in this study. The demographic data of the subjects is summarized in Table 1. The mean age of the patients was $69.5 \pm 12.5$ years, ranging from 30 to 94 years. The clinical profiles of NSCLC patients were adenocarcinoma (28/44; 63.64%) and squamous cell carcinoma (16/44; 36.36%). Most lung adenocarcinoma patients were female (19/28; 67.86%). Male patients were diagnosed with lung squamous cell carcinoma more often than females (15/16; 93.75%). The stage at the time of diagnosis was determined according to the tumor, node and metastasis (TNM) staging system, as shown in Table 1. There were three lung cancer patients (one from each category: adenocarcinoma, squamous cell carcinoma, and SCLC) for whom we were unable to obtain their TNM stages. In comparison with NSCLCs, most SCLC patients had significantly poor prognosis due to factors such as tumor size ($6.3 \pm 3.1$ cm.; $p = 0.009$), lymph node metastasis (85.00% of all SCLC cases; $p < 0.001$), distant metastasis (75.00% of all SCLC cases; $p < 0.001$), and late stage (III–IV) (94.74% of all SCLC cases; $p < 0.001$).

## G6PD immunostaining

To examine whether G6PD expression could be correlated with the tumorigenesis of lung tissues, we performed immunohistochemistry of G6PD protein in lung tissues obtained

**Table 1  Summary of patient characteristics.**

| Clinical parameters | Total n = 64 (%) | NSCLC n = 44 (%) | SCLC n = 20 (%) | p-value[a] | NSCLC | | |
| --- | --- | --- | --- | --- | --- | --- | --- |
| | | | | | Adeno-carcinoma n = 28 (%) | Squamous cell carcinoma n = 16 (%) | p-value[b] |
| Males | 40 (62.5) | 24 (54.5) | 16 (80.0) | 0.051 | 9 (32.1) | 15 (93.8) | <0.001 |
| Age (Years) (Mean ± SD) | 69.5 ± 12.5 | 70.1 ± 13.2 | 68.2 ± 10.8 | 0.577 | 68.2 ± 11.4 | 73.4 ± 15.8 | 0.208 |
| Tumor size (cm.) (Mean ± SD) | 4.6 ± 2.6 | 4.0 ± 2.1 | 6.3 ± 3.1 | 0.009 | 3.2 ± 1.2 | 5.3 ± 2.7 | 0.001 |
| Lymph node metastasis | 33 (51.6) | 16 (36.4) | 17 (85.0) | <0.001 | 11 (39.3) | 5 (31.3) | 0.594 |
| Distant metastasis | 29 (45.3) | 14 (31.8) | 15 (75.0) | <0.001 | 11 (39.3) | 3 (18.8) | 0.172 |
| TNM | | | | | | | |
| Early stage (I, II) | 61 (95.3) | 27 (64.3) | 1 (5.3) | <0.001 | 19 (70.4) | 8 (53.3) | 0.270 |
| Advance stage (III, IV) | | 15 (35.7) | 18 (94.7) | | 8 (29.6) | 7 (46.7) | |
| G6PD IHC positive | 38 (59.3) | 36 (81.8) | 2 (10.0) | <0.001 | 24 (85.7) | 12 (75.0) | 0.434[c] |
| H-score (Mean ± SD) | 40.7 ± 31.9 | 41.5 ± 32.6 | 25.8 ± 1.1 | 0.007 | 30.3 ± 23.2 | 63.9 ± 37.9 | 0.013 |
| Recurrent | 17 (26.6) | 13 (29.5) | 4 (20.0) | 0.315 | 9 (32.1) | 4 (25.0) | 0.171 |
| Death | 12 (18.8) | 7 (15.9) | 5 (25.0) | 0.057[c] | 3 (10.7) | 4 (25.0) | 0.343[c] |

Notes.
[a]Compared between NSCLC & SCLC.
[b]Compared between adenocarcinoma and squamous cell carcinoma.
[c]Fisher's exact test.

from lung cancer patients. Results revealed that the G6PD protein was highly expressed in cancerous areas of lung tissues compared to adjacent normal tissues (Fig. 1). The staining was more intense in NSCLC than in SCLC tissues ($p < 0.001$) (Fig. 1, Table 1). Consistently, the H-scores of G6PD expression were higher in NSCLC than in SCLC tissues ($p = 0.007$) (Table 1). Among NSCLC tissues, the H-scores of G6PD in squamous cell carcinoma were higher than those in adenocarcinoma tissues ($p = 0.013$) (Table 1).

## G6PD and NSCLC cell proliferation

In this set of experiments, we aimed to investigate a correlation between G6PD expression and NSCLC growth. The growth of three cell lines with different characteristics was compared. The first cell line was Beas-2B, epithelial cells isolated from normal human bronchial epithelium obtained from autopsies of noncancerous individuals. The second cell line, NCI-H292, was derived from pulmonary mucoepidermoid carcinoma, whereas the third cell line, NCI-H1975, was lung epithelial cells derived from adenocarcinoma tissue. All cells were seeded at the same cell density. After 48 h, the MTT cell proliferation assay was performed to quantify the number of viable cells. Results showed that NCI-H292 cells had a higher proliferation rate than NCI-H1975 and Beas-2B cells (Fig. S1). Then, we compared the mRNA expression levels and activity of G6PD in all cell lines. Results showed that mRNA expression levels and activity of G6PD were significantly higher in NCI-H292 cells than in NCI-H1975 and Beas-2B cells (Fig. S1). These findings suggested that expression levels and activity of G6PD had a positive correlation with the growth of NSCLC cells.

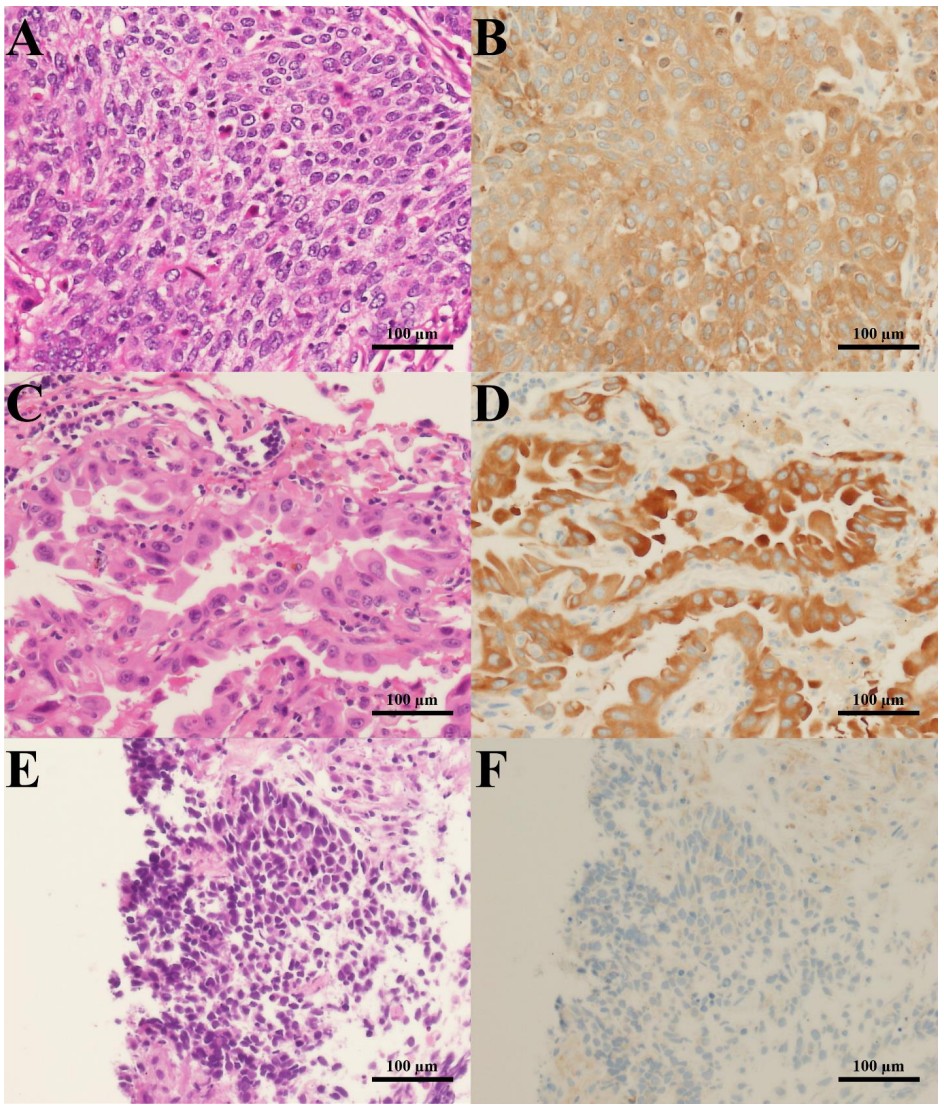

**Figure 1** **H&E staining and G6PD IHC staining of lung cancer tissues.** Representative images of H&E staining (left column) and G6PD IHC staining (right column) (200 × magnification) (scale bar = 100 μm) of lung cancer tissues from patients with (A, B) squamous cell carcinoma, (C, D) adenocarcinoma, and (E, F) SCLC.

## Inhibition of G6PD reduced NSCLC cell proliferation

We challenged the role of G6PD in the growth of NSCLS cells by using DHEA, a known G6PD inhibitor (*Preuss et al., 2013*), and small interfering RNA for G6PD (siG6PD). Results in the first set of experiments with DHEA demonstrated that, after 48 h of treatment, DHEA exhibited a concentration-dependent reduction of G6PD activity in both NCI-H292 and NCI-H1975 cells (Figs. 2A–2B). Although DHEA at 400 μM showed superior results in the reduction of G6PD activity in both NSCLC cells, as compared to DHEA at 300 μM, we observed its incomplete solubility at this concentration. Therefore, we selected DHEA

at the concentration of 300 µM for the evaluation of NSCLC cell proliferation using MTT and colony forming assays. After 48 h of treatment, DHEA significantly reduced cell proliferation of both NSCLC cells (Fig. 2C). If NSCLC cells were allowed to grow in normal culture conditions for 7 days after DHEA treatment, there were no colonies observed at the end of the experiment compared to corresponding controls (Fig. 2D) in NCI-H1975 (Figs. 2E–2F, Fig. S2) and NCI-H292 (Figs. 2G–2H, Fig. S2).

In the next set of experiments, siG6PD was used to suppress G6PD expression in NSCLC cells. The efficacy of siG6PD after 24 h of application was evaluated by measuring G6PD mRNA levels and G6PD activity. Results showed that siG6PD significantly reduced G6PD expression and G6PD activity in NCI-H1975 and NCI-H292 cells by more than 50% compared to scramble controls (Figs. 2I–2J). Given that the G6PD mRNA levels in NCI-H1975 cells were significantly lower than those in NCI-H292 cells (Fig. S1), the knockdown of G6PD mRNA using siG6PD at the same concentration (100 nM) had a more pronounced effect on G6PD mRNA expression levels in NCI-H1975 cells compared to NCI-H292 cells. After 24 h of siG6PD application, the cell proliferation of NSCLC cells was determined. Results showed that siG6PD significantly reduced cell proliferation of both NCI-H1975 cells and NCI-H292 cells by approximately 20% compared to scramble controls (Fig. 2K).

### Inhibition of G6PD affected NSCLC cell migration.

In this set of experiments, a scratch-wound assay was performed 24 h post-treatment to assess the effects of G6PD inhibition on the contribution of NSCLC cell migration to wound closure. Results demonstrated that DHEA at 300 µM decreased wound confluence by increasing unmigrated area in cultures of NCI-H1975 and NCI-H292 cells, compared with that of the untreated controls at every time point examined (Figs. 3A–3D, Figs. S3 and S4). Initially, an application of siG6PD slightly decreased wound confluence in cultures of NCI-H292 cells compared with that of scramble controls. However, its effect was significantly observed at 24 h post-application (Figs. 3F and 3H, Fig. S5). In contrast, an application of siG6PD showed less effectiveness in decreasing wound confluence in cultures of NCI-H1975 cells at all time points examined, when compared to scramble controls (Figs. 3E and 3G, Fig. S6).

### DHEA and siG6PD induced NSCLC cell apoptosis

To investigate whether the mechanism underlying the inhibitory effects of G6PD on NSCLC cell proliferation involved the induction of apoptosis, we measured the levels of the Bax/Bcl-2 mRNA ratio. Results indicated that the reduction of G6PD activity after 48 h of DHEA treatment and the downregulation of G6PD expression by siG6PD after 24 h post-application significantly induced apoptosis in NCI-H292 cells by increasing the Bax/Bcl-2 mRNA ratio when compared to corresponding controls. However, the same treatments did not induce significant changes in the Bax/Bcl-2 mRNA ratio in NCI-H1975 cells (Figs. 4A–4B).

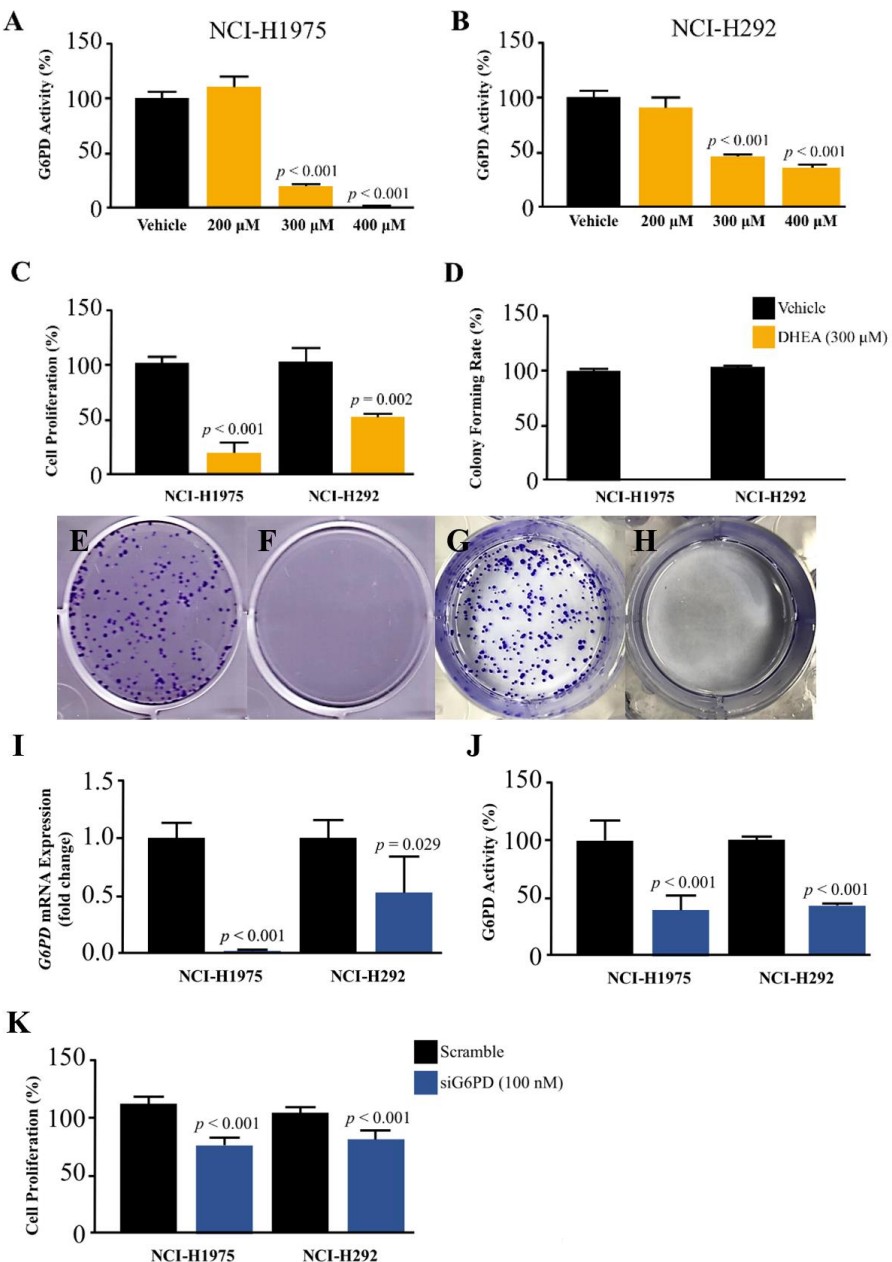

**Figure 2   Effect of DHEA and siG6PD on NSCLC cell proliferation.** Inhibitory effects of DHEA on G6PD activity in (A) NCI-H1975 cells and (B) NCI-H292 cells, (C) cell proliferation in NCI-H1975 cells and NCI-H292 cells at 48 h, (D) colony forming rate and colony formation in NCI-H1975 (E and F) and NCI-H292 (G and H) on day 7 after 48 h of DHEA treatment in vehicle and DHEA, respectively. Inhibitory effects of siG6PD after 24 h of application on (I) G6PD mRNA levels, (J) G6PD activity, and (K) cell proliferation of NCI-H1975 and NCI-H292 cells. Data are presented as mean ± SD ($n = 3$), and significant differences are indicated as $p < 0.05$.

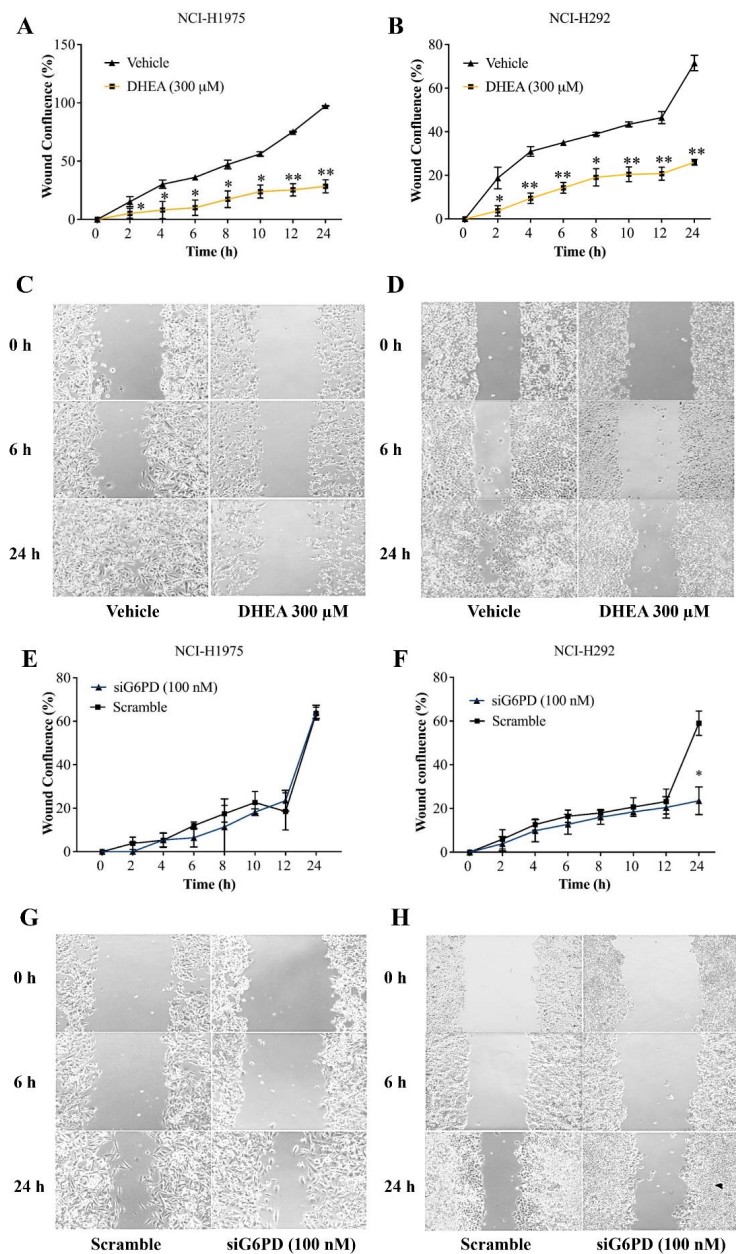

**Figure 3** **Inhibitory effects of DHEA and siG6PD on wound confluence of NSCLC cells in a scratch-wound assay over a 24-hour period as indicated.** Percentage of wound confluence (A and B) in cultures of NCI-H1975 cells and of NCI-H292 cells, and (C and D) representative images of the assay in cultures of NCI-H1975 cells and of NCI-H292 cells in the presence of 300 μM DHEA compared to that of corresponding vehicle controls. Percentage of wound confluence (E and F) in cultures of NCI-H1975 cells and of NCI-H292 cells, and (G and H) representative images of the assay in cultures of NCI-H1975 cells and of NCI-H292 cells after siG6PD application compared to that of corresponding scramble controls. Data are presented as mean ± SD ($n = 3$). Significant differences compared to vehicle or scramble (in the treatment of DHEA or siG6PD, respectively) were indicated as *$p < 0.05$ and **$p < 0.001$.

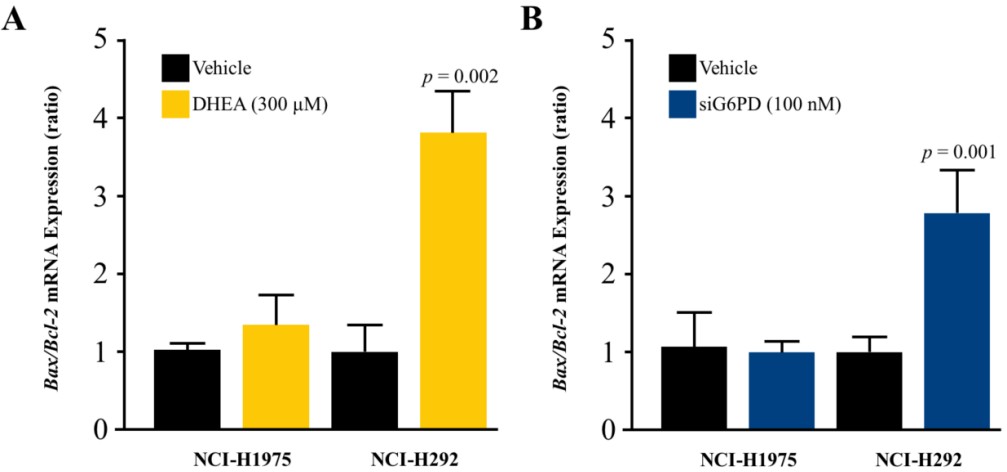

**Figure 4** **Effect of DHEA and siG6PD on NSCLC cell apoptosis.** Effects of (A) DHEA at 48 h post-treatment and (B) siG6PD at 24 h post-application on the levels of Bax/Bcl-2 mRNA ratio in NCI-H1975 and NCI-H292 cells. Data are presented as mean ± SD ($n = 3$), and significant differences are indicated as $p < 0.05$.

## Co-treatment effects of D-(-)-ribose and DHEA/siG6PD on NSCLC cell proliferation

To highlight the role of PPP in NSCLC cell proliferation, we asked whether the addition of D-(-)-ribose, an end-product of PPP (*Mahoney et al., 2018*), could reverse the inhibitory effects of DHEA and siG6PD on NSCLC cell proliferation. In this set of experiments, D-(-)-ribose was co-administered with either DHEA or siG6PD. Results demonstrated that addition of D-(-)-ribose did not alter cell proliferation of NCI-H1975 cells that were treated with DHEA or siG6PD, compared to DHEA or siG6PD-treated controls (Figs. 5A–5B). In contrast, D-(-)-ribose increased cell proliferation of NCI-H292 cells treated with DHEA or siG6PD to reach the levels of untreated control cells in a concentration-dependent manner (Figs. 5A–5D).

## DISCUSSION

G6PD is the rate-limiting enzyme in PPP and has been linked to the tumorigenesis of many types of cancer, including lung cancer. Here, we reported a positive correlation between high expression levels of G6PD and NSCLC. This finding was in line with the CU-DREAM analysis of NCBI's GEO dataset program (*Aporntewan & Mutirangura, 2011*), targeting the overexpression of G6PD protein in lung cancer tissues compared to adjacent normal tissues. Furthermore, the results from a series of *in vitro* experiments indicated the important role of G6PD in NSCLC cell proliferation. These findings have implications for targeting the role of G6PD in carcinogenesis and the development of novel strategies for lung cancer therapy.

Most lung cancer patients in this study were elderly, with a mean age of 69.5 ± 12.5 years, similar to other studies (*Blanco et al., 2015*). This can be explained by a

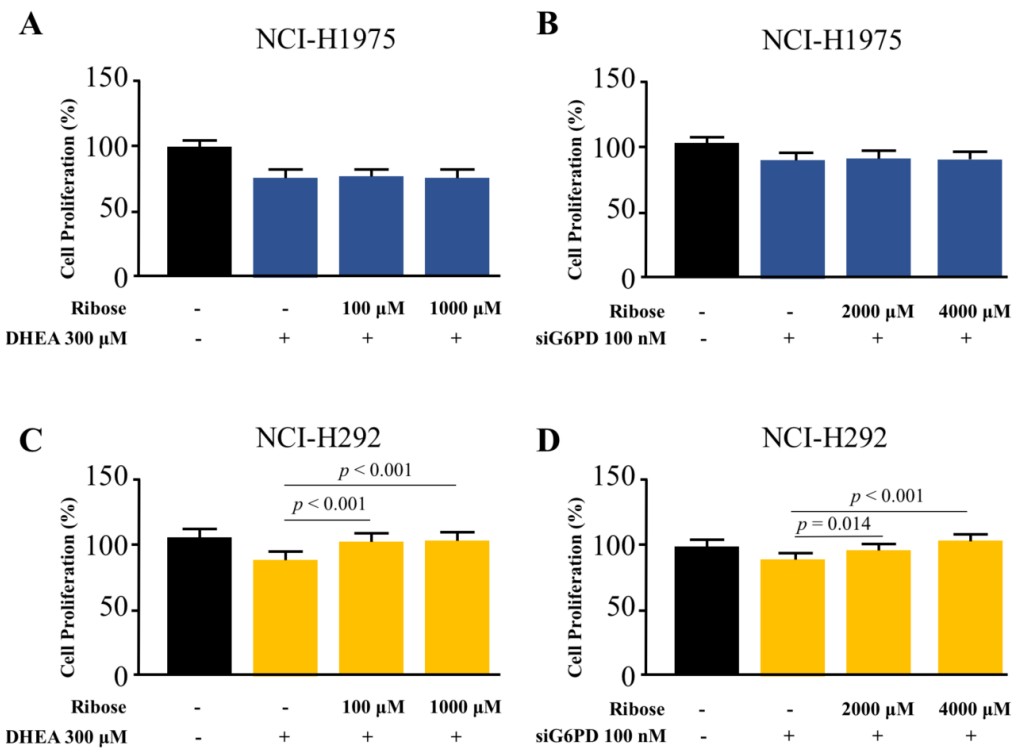

**Figure 5 Effects of co-treatment of D-(-)-ribose and DHEA/siG6PD on NSCLC cell proliferation.** MTT assay was performed to evaluate cell proliferation of NCI-H1975 cells (A and B) and NCI-H292 cells (C and D) that were co-treated with D-(-)-ribose and DHEA for 48 h or D-(-)-ribose and siG6PD for 24 h. Data are presented as mean ± SD ($n = 3$), and significant differences are indicated as $p < 0.05$.

number of risk factors, including delayed diagnosis, erroneous staging, a lack of effective screening methods, and patient behaviors, including smoking habits, prolonged exposure to carcinogens, and the duration of carcinogenesis (*Ellis & Vandermeer, 2011*; *Venuta et al., 2016*). NSCLC was found to be the most common subtype. In agreement with previous reports, male patients in this study were more likely to be affected by squamous cell carcinoma (*Pesch et al., 2012*; *Venuta et al., 2016*). On the other hand, our female patients had a higher frequency of adenocarcinomas.

This present study reported for the first time that G6PD was highly expressed in NSCLC tissues compared to SCLC tissues. G6PD expression was more intense in squamous cell carcinoma than adenocarcinoma. According to the American Type Culture Collection (ATCC) cell line information, the NCI-H292 cell line exhibits various markers of squamous differentiation, whereas the NCI-H1975 cells originate from adenocarcinoma tissue. This information is in line with our findings of high levels of G6PD activity and mRNA expression in the NCI-H292 cell line compared to the NCI-H1975 cell line.

The variation in G6PD levels between these two NSCLC cell lines exhibited a positive correlation with their cell type and proliferation rates, emphasizing the significance of G6PD in the specific cell type and growth of NSCLC. Our study is limited by a small sample

size and a lack of detailed information regarding the specific treatments received by the patients.

Our findings that DHEA treatment and application of siG6PD reduced cell proliferation and colony formation in NSCLC cells were supported by previous studies. *Fang et al. (2016)* showed that DHEA and G6PD shRNA decreased the viability of cervical cancer cells (*Fang et al., 2016*). DHEA treatment suppressed the colony formation of hepatocarcinoma and breast cancer cells (*Cheng et al., 2016*; *Lopez-Marure et al., 2016*). Therefore, targeting G6PD might be of therapeutic benefit for several types of cancer.

The present study demonstrated that DHEA treatment and siG6PD application reduced NSCLC cell migration in a scratch-wound assay. These findings were supported by several studies in cervical cancer (*Fang et al., 2016*), oral squamous cell carcinoma (*Wang et al., 2020*), and breast cancer (*Lopez-Marure et al., 2016*). The underlying mechanisms might involve the suppression of epithelial-mesenchymal transition (EMT) through E-cadherin activation (*Wang et al., 2020*). Additionally, we demonstrated that inhibition of G6PD activity and down-regulation of G6PD mRNA expression elevated the Bax/Bcl-2 mRNA ratio, indicating an increasing susceptibility of NSCLC cells to apoptosis. A distinction in characteristics was observed between these two cell lines. Specifically, cell proliferation, mRNA expression levels, and the activity of G6PD were significantly higher in NCI-H292 cells compared to NCI-H1975 cells. These findings suggest that NCI-H292 cells rely on G6PD for their accelerated growth. Consequently, inhibiting G6PD activity and reducing G6PD mRNA expression in NCI-H292 cells increased their susceptibility to apoptosis more than that of NCI-H1975 cells. DHEA and siG6PD could deplete two main products of G6PD, NADPH and ribose, *via* the PPP. Low levels of NADPH could increase the susceptibility of NSCLC to oxidative stress. When G6PD was inhibited, ROS and redox imbalances could occur, leading to ROS-mediated apoptosis (*Yang et al., 2019*). Simultaneously, reduced levels of ribose could compromise DNA synthesis in NSCLC cells, leading to cell cycle arrest and apoptosis, respectively. These mechanisms could be used to explain the reduction in wound confluence in cultures of NSCLC cells treated with DHEA and siG6PD.

In order to study the proliferative effect in combination with DHEA or siG6PD on G6PD inhibition, D-(-)-ribose, the precursor for nucleic acid synthesis, has been used as a supplement. In G6PD repressed NSCLCs, D-(-)-ribose reversed the antiproliferative effects for the first time, as shown in our study. However, NCI-H292 was able to re-proliferate at low concentrations of D-(-)-ribose after being inhibited by DHEA, whereas a high concentration was needed for the inhibition of G6PD mRNA *via* siG6PD. As mentioned in NCI-H1975, previous studies also demonstrated the incapability of D-(-)-ribose to rescue the growth of G6PD knockdown cells, possibly due to the insufficient level of G6PD in ROS scavenging and ribose synthesis under oxidative stress or DNA damage (*Xu et al., 2016*). Different concentrations of D-(-)-ribose have been used in various cancer cell types differently (*Croci et al., 2011*; *Xu et al., 2021*), possibly due to the nature, aggressiveness, and baseline G6PD activity level of each lung cancer cell. As stated in a previous study, differential regulation of metabolic pathways in different NSCLC subtypes may contribute to different metabolic vulnerabilities, which can be indicated as potential therapeutic

targets (*Sellers et al., 2019*). A chemotherapy protocol centered around the use of cisplatin is a commonly utilized approach in the treatment of advanced NSCLC (*Hong et al., 2018*). In the cisplatin-resistant model, G6PD mRNA and protein expression, along with glutathione and ROS levels, significantly increase, highlighting the link between G6PD and cisplatin-resistant lung cancer. Notably, G6PD inhibition has a more pronounced impact on cisplatin-resistant lung cancer cells compared to sensitive ones (*Hong et al., 2018*). For this reason, G6PD could be a potential therapeutic target for NCI-H292 cell lines. Taken together, these can be stated to indicate that suppression of G6PD by DHEA or siG6PD altered EMR through the regulation of PPP in lung cancer cell lines. Further studies will delve into the effects of G6PD in a cisplatin resistance model and *in vivo* investigations while monitoring redox homeostasis.

## CONCLUSIONS

Our findings demonstrated for the first time that G6PD is predominantly expressed in NSCLC and clarified the key roles of G6PD in lung cancer cell proliferation and apoptosis through the regulation of PPP and nucleotide synthesis. Therefore, G6PD could be a potential therapeutic strategy for lung cancer treatment.

## ACKNOWLEDGEMENTS

The authors extend our gratitude with special thanks to Prof. Dr. Apiwat Mutirangura for generously providing the cell lines NCI-H1975 and NCI-H292, and Prof. Dr. Pithi Chanvorachote for supplying the Beas-2B cell lines.

### Funding

This study was supported by the 100th Anniversary Chulalongkorn University Fund for Doctoral Scholarship, the 90th Anniversary of Chulalongkorn University Fund (Ratchadaphiseksomphot Endowment Fund), Ratchadapiseksompotch Fund, Faculty of Medicine, Chulalongkorn University (Grant No. 065/24) (Grant No. RA61/039) and the Thailand Government Budget Fund 2018 (Ratchadaphiseksomphot Endowment Fund) (Grant No. 1GB_CU_61_22_30_12). The funders had no role in study design, data collection and analysis, decision to publish, or preparation of the manuscript.

### Grant Disclosures

The following grant information was disclosed by the authors:
100th Anniversary Chulalongkorn University Fund for Doctoral Scholarship.
90th Anniversary of Chulalongkorn University Fund (Ratchadaphiseksomphot Endowment Fund).
Ratchadapiseksompotch Fund.
Faculty of Medicine, Chulalongkorn University: 065/24, RA61/039.
Thailand Government Budget Fund 2018 (Ratchadaphiseksomphot Endowment Fund): 1GB_CU_61_22_30_12.

## Competing Interests

The authors declare there are no competing interests.

## Author Contributions

- Makamas Chanda performed the experiments, analyzed the data, prepared figures and/or tables, authored or reviewed drafts of the article, and approved the final draft.
- Pornchai Anuntasomboon performed the experiments, analyzed the data, prepared figures and/or tables, and approved the final draft.
- Komkrit Ruangritchankul performed the experiments, prepared figures and/or tables, authored or reviewed drafts of the article, and approved the final draft.
- Poonlarp Cheepsunthorn conceived and designed the experiments, authored or reviewed drafts of the article, and approved the final draft.
- Chalisa L. Cheepsunthorn conceived and designed the experiments, authored or reviewed drafts of the article, and approved the final draft.

## Human Ethics

The following information was supplied relating to ethical approvals (*i.e.*, approving body and any reference numbers):

This study was approved by the Institute Ethics Committee of the Faculty of Medicine, Chulalongkorn University, Bangkok, Thailand (IRB 561/59, COA No. 1034/2016).

## Data Availability

The raw measurements are available in the Supplementary File.

## Supplemental Information

Supplemental information for this article can be found online at http://dx.doi.org/10.7717/peerj.16503#supplemental-information.

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
