# Peer review of "Inhibition of non-small cell lung cancer (NSCLC) proliferation through targeting G6PD"

_PeerJ, doi:10.7717/peerj.16503_

## Round 0.1 · original submission · Major Revisions

Please revise and resubmit at the earliest.

Reviewer 1 ·

Basic reporting

This study explores the expression level of G6PD in lung tissue samples from 64 patients and its role in Non-Small Cell Lung Cancer (NSCLC) cell lines in the context of cell apoptosis. The analysis of clinical data is comprehensive and likely to stimulate substantial research interest. Additionally, the paper is well-structured, providing a smooth reading experience.

Experimental design

However, most of the findings only stem from in vitro cell line studies.

Validity of the findings

For a more compelling perspective, I recommend integrating ex vivo lung cancer tissues or an in vivo NSCLC mouse model. This would enhance the validity of G6PD as a therapeutic target, as evaluated using DHEA or D-(-)- ribose.

Additional comments

The following specific concerns need to be addressed.

Annotated reviews are not available for download in order to protect the identity of reviewers who chose to remain anonymous.

·

Basic reporting

The manuscript entitled "Inhibition of non-small cell lung cancer (NSCLC) proliferation through targeting Glucose 6-phosphate dehydrogenase{G6PD}described the growth-promoting effects of G6PD in non- small cell lung cancer (NSCLC)..

Experimental design

Clinical characteristics and G6PD expression levels in lung tissues of 64 patients
diagnosed with lung cancer at the King Chulalongkorn Memorial Hospital (Bangkok, Thailand)
during 2009-2014 were analyzed. G6PD activity in NSCLC cell lines, including NCI-H1975 and
NCI-H292, was experimentally inhibited using DHEA and siG6PD to study cancer cell proliferation and migration.
DHEA{Dehydroepiandrosterone}, an adrenal steroid is a potent uncompetitive inhibitor of mammalian glucose-6-phosphate dehydrogenase (G6PDH). Following methods have been used to meet the objective of the study.
{A.}Tissue specimens :-Lung tissue specimens were obtained from 64 lung cancer patients who were
admitted to the King Chulalongkorn Memorial Hospital (Bangkok, Thailand) during 2009-2014.
All tissue samples were examined by a team of pathologists at the King Chulalongkorn
Memorial Hospital to determine the type and stage of cancer.
{B.}Histological and immunohistochemical analyses
{C.}Cell culture and treatments :- Human NSCLC cell lines, NCI-H1975 (lung epithelial cells derived
from adenocarcinoma tissue; ATCC# CRL-5908) and NCI-H292 (a lymph node metastasis of
a pulmonary mucoepidermoid carcinoma; ATCC# CRL-1848), were obtained from the
American Type Culture Collection (ATCC, Manassas, VA, USA)..
{D.}siG6PD and transfection assay
{E.}Cell proliferation assay.
{F.}Clonogenic assay
{G}G6PD activity assay.
{H.}Scratch wound assay
{I.}Quantitative RT-PCR
{J}Statistical analysis
All the experiments were obtained from three independent experiments performed in triplicate and presented as the mean ± standard deviation (SD). The chi-square test (Ç2) and Student' t-test were used to examine the differences between categorical and quantitative variables. A two-sided difference
with a p-value less than 0.05 was considered statistically significant.

Validity of the findings

{A.}The investigators claim that the present study reported for the first time the high expression of G6PD in NSCLC {non-small cell lung cancer} tissues as compared to SCLC {small cell lung cancer} tissues.
{B.}G6PD{Glucose 6-phosphate dehydrogenase} expression was more intense in squamous
cell than adenocarcinoma. This finding was in line with the CU-DREAM analysis of NCBIís
GEO dataset program, targeting the over expression of G6PD protein in lung cancer tissues
compared to adjacent normal tissues.
{C.} Their study also demonstrated a positive correlation between the advanced stages of NSCLC
and G6PD over expression,in tune with previous reports of high levels of G6PD expression
in poor prognosis lung cancer patients
{D.} Findings from this investigation have implications for targeting the role of G6PD in carcinogenesis
and the development of novel strategies for lung cancer therapy.
{E.}The differences in G6PD levels in these two NSCLC cell lines{NCI-H292,NCI- H1975 cell}
positively correlated with their cell proliferation rates, suggesting the importance of G6PD
in the growth and advancement of NSCLC cells.
{F.}DHEA treatment and application of siG6PD reduced cell proliferation and colony formation in
NSCLS cells were in tune with the published literature.
{G.}The present study demonstrated that DHEA treatment and siG6PD application reduced NSCLC
cell migration in a scratch-wound assay which concurs with earlier published literature.
The authors is of the opinion that the underlying mechanisms might involve the suppression
of epithelial-mesenchymal transition(EMT) through E-cadherin activation.
{H} The authors have demonstrated that inhibition of G6PD activity and down-regulation of G6PD
mRNA expression elevated the Bax/Bcl-2 mRNA ratio, indicating an increasing susceptibility of
NSCLC cells to apoptosis.
{I} Hence, G6PD could be a potential therapeutic target for NCI-H292 cell lines.The authors stated
that the results indicate that suppression of G6PD by DHEA or siG6PD altered EMR through
the regulation of PPP in lung cancer cell lines.

Additional comments

Nil

Reviewer 3 ·

Basic reporting

Overall this is a well-written manuscript. However, I suggest authors consider comments below and in the next sections:

Please explain the rationale for this study well.

Experimental design

Cisplatin-based interventions are s a commonly used chemotherapy regimen in advanced patients. I suggest the authors look into how response/sensitivity towards Cisplatin is influenced by Inhibition of G6PD

I suggest authors also include additional markers of cellular redox homeostasis to make this study robust and further useful for the field.

Without proper controls, G6PD H-score calculations and resultant binning is not very helpful.

Sample selection and analysis:
Did these patients undergo any prior medical treatment? Evaluation and analysis of what types of treatments patients undergo before sample collection would be an important factor affecting metabolic programming. I suggest authors evaluate details of medical history, and rebin patients, and provide insights into how analyses were controlled.

I suggest authors also look at other enzymes upstream of G6PD to determine if this is a pathway-specific pattern.

Figure 2 (corresponding experiments) needs control samples.

Mention in figure legends: How many samples? How many repeats?

How do authors explain G6PD mRNA levels in NCI-H1975 and NCI-H292 Figure 4A?

Validity of the findings

Several more controls are needed to be included in the study to make findings robust.

---

## Round 0.2 · Minor Revisions

Please revise the figures and resubmit at the earliest.

Reviewer 1 ·

Basic reporting

The length of the scale bars is not specified. Please update the figure/legends by adding the corresponding scale bar lengths. Then the manuscript will be in a good format and ready for publication.

Experimental design

no comment

Validity of the findings

no comment

·

Basic reporting

Comments of the reviewers 1 and 3 were answered and their suggestions were incorporated in the modified manuscript.

Experimental design

The authors have followed well defined methods systematically and carried out the experiments.

Validity of the findings

Already mentioned with respect to the original manuscript.

Additional comments

Nil

---

## Round 0.3 · accepted · Accept

Congratulations, your paper has been accepted.